# The Effect of Different Mixed Organic Solvents on the Properties of p(OPal-MMA) Gel Electrolyte Membrane for Lithium Ion Batteries

Lanlan Tian [1,2], Mengkun Wang [1,3,4,5], Lian Xiong [1,3,4,5], Haijun Guo [1,3,4,5], Chao Huang [1,3,4,5], Hairong Zhang [1,3,4,5,*] and Xinde Chen [1,3,4,5,*]

[1] Guangzhou Institute of Energy Conversion, Chinese Academy of Sciences, Guangzhou 510640, China; plshm126@126.com (L.T.); wangmengkun_0715@163.com (M.W.); xionglian@ms.giec.ac.cn (L.X.); guohaijun6621@163.com (H.G.); huangchao@ms.giec.ac.cn (C.H.)
[2] University of Chinese Academy of Sciences, Beijing 100049, China
[3] Key Laboratory of Renewable Energy, Chinese Academy of Sciences, Guangzhou 510640, China
[4] Guangdong Provincial Key Laboratory of New and Renewable Energy Research and Development, Guangzhou 510640, China
[5] R&D Center of Xuyi Attapulgite Applied Technology, Guangzhou Institute of Energy Conversion, Chinese Academy of Sciences, Xuyi 211700, China
* Correspondence: hairongz@gmail.com (H.Z.); cxd_cxd@hotmail.com (X.C.); Tel./Fax: +86-20-3721-3916 (H.Z. & X.C.)

**Abstract:** A solvent is a key factor during polymer membrane preparation, and it is directly related to application performance as a separator for lithium ion battery (LIB). In this study, different mixed solvents were employed to prepare polymer (p(OPal-MMA)) membranes by the phase inversion technique. The polymer membrane then absorbed liquid electrolytes to obtain gel electrolytes (GPEs). The surface morphologies and porosities of these membranes were investigated, and lithium ion transferences and electrochemical performances of these GPEs were also measured. The membrane displayed an interconnected three-dimensional framework structure with uniformly distributed pores when using DMF as a porogen. When combined with acetone as the component solvent, the prepared GPE displayed the largest lithium ion transference number (0.706), the highest porosity (42.6%) and ion conductivity ($3.99 \times 10^{-3}$ S/cm). Even when assembled as Li/GPE/LiFePO$_4$ cell, it exhibited the highest initial specific capacity of 167 mAh/g and retained most capacity (162 mAh/g) after 50 cycles. The results presented here probably provide reference for choosing an appropriate mixed solvent in fabricating polymer membranes.

**Keywords:** lithium ion battery; phase inversion; mixed solvent; gel polymer electrolyte

## 1. Introduction

Lithium ion batteries (LIBs), as a new kind of energy storage devices, has been receiving growing attention especially due to its high-energy density, high operating voltage, and long cycling life, among others [1–3]. LIBs have been extensively applied in portable electronic devices, electric vehicles and energy storage systems [4,5]. The membrane is a significant part of LIB, playing an important role in preventing an internal short circuit and ensuring smooth lithium ion migration between the cathode and the anode [4,6]. The common commercial membrane materials are mainly polyolefin, such as polypropylene (PP) and polyethylene (PE), due to their good mechanical properties and electrochemical stabilities [4,7]. However, the polyolefin membrane has some drawbacks such as low electrolyte uptake, poor electrolyte retention ability, and inability to absorb electrolyte with high dielectric constants, thus

resulting in a low ionic conductivity. To enhance the ionic conductivity of the membrane, many polymer electrolyte membranes (copolymers, block copolymers and random copolymers) are widely studied as a promising membrane applied in LIB because they act not only as an ionic conductor but also as a membrane [8–12]. Among these polymer electrolyte membrane, poly(methyl methacrylate) (PMMA), a kind of gel polymer electrolyte (GPE) matrix, has been considered as a promising membrane candidate, owing to its high electrolyte uptake and good compatibility with electrode materials [13].

Although PMMA as the matrix of GPE has been studied extensively, there are still some problems including low ionic conductivity and poor mechanical integrity etc., which greatly limit its development in LIB. Various nano particles such as $SiO_2$, $TiO_2$ and clay have been used for modifier in the PMMA matrix membrane in order to enhance the properties of PMMA [13–15]. Palygorskite (Pal) is a kind of fibrous magnesium aluminosilicate clay. Pal has many distinctive advantages as a modifier, including appropriate interlayer charge, large specific surface area, high aspect ratio and high cation-exchange capacity. Chen et al. [16] found that the Pal can enhance prominently thermal stability and mechanical property of PMMA after Pal was modified by 2,4-tolylene diisocyanate. In the previous study [17], a inorganic-organic hybrid polymer (poly(organic Pal-co-methyl methacrylate), p(OPal-MMA)) was prepared from MMA and organic Pal. The polymer has many net shaped microporou structures and the ionic conductivity of the hybrid polymer-based GPE was $2.94 \times 10^{-3}$ S/cm. However, during experiments, we found that the preparation technologies of the gel polymer membrane using different organic solvents also affect electrochemical characters. Thus, in this study, the preparation technology of GPE is reported especially by phase inversion method using different mixing solvents.

On the other hand, many studies have been focused on preparation of the PMMA membrane using different membrane forming methods, such as Bellcore technique [18], solvent casting [19,20], electrospun [7,21] and phase inversion [22,23]. The Bellcore technique has always been utilized in preparing porous polymer membranes with high ionic conductivity at room temperature [18]. However, the extraction process consumes a great deal of organic solvent, largely raising production costs. Solvent casting uses a single solvent (THF or dimethyl sulfoxide, DMSO) to dissolve the polymer and prepare the membrane, but the pores of the membrane are rare and ionic conductivity is rather low [19,20]. The electrospun technique is also not significant in improving ionic conductivity performance [7,21]. Compared with the above techniques, the phase inversion method is easier and makes pore formation in the membrane more convenient. A competitive mutual diffusion thus takes place between the solvent and non-solvent, resulting in a uniform porous structure [24–26]. In particular, the porosity of the polymer membrane can be controlled by varying the kind, content and separative condition of the organic solvent to form the membrane with a uniform porous structure [27]. Moreover, the controllable finished shape of the polymer membrane and the simple preparation process greatly contribute to realizing largescale industrial production. Xiao et al. [28] prepared a novel macro-porous polymer electrolyte based on PVDF/PEO (poly(ethylene oxide))-b-PMMA using the phase inversion technique. They found that the PEO-b-PMMA block copolymer could obviously optimize the porous structure and improve connectivity [28]. Generally, the solvents in the phase-inversion method are tetrahydrofuran (THF) [29], dimethylformamide (DMF) [30], and acetone [31] among others. However, the low ionic conductivity of the membrane is occurred when employing single solvents like THF or DMF, combined with off-size pores.

The organic solvent is a key that affects the structure and electrochemical properties of the p(OPal-MMA) polymer membrane; however, there is little study on it. Thus, the preparation technology of GPE by phase inversion method was investigated in this study. The bi-solvent two-step phase-inversion method was applied to prepare a uniform porous polymer membrane and the pore volume and pore size of the membrane were controlled by changing the ratio of the different solvents, and hence the liquid electrolyte uptake and ionic conductivity of the membrane could be controlled as well. We used p(OPal-MMA) as material to prepare the self-supporting microporous p(OPal-MMA) membrane. The mixed organic solvents were composed of porogen (DMF or DMAC) and solvent

(acetone or THF) with different concentrations. The corresponding physical and electrochemical characters of p(OPal-MMA) membrane was also investigated.

## 2. Methods

### 2.1. Materials

1-methyl-2-pyrrolidinon (NMP), tetrahydrofuran (THF), *N,N*-Dimethylacetamide (DMAC), acetone and *N,N*-Dimethylformamide (DMF) were purchased from Kermel (TianJing, China). Carbon black and lithium iron phosphate (LiFePO$_4$) powder were supplied by Beijing HWRK Chem Co. Ltd., (Beijing, China). Binder (polyvinylidene fluoride, PVDF), lithium tablets and aluminum foil were obtained from Mingruixiang Automation Equipment Co., Ltd. (Shenzhen, China).

### 2.2. Preparation Process of the p(OPal-MMA) Polymer Membranes and GPE

The p(OPal-MMA) was synthesized by solution polymerization according to a previous study [17]. The p(OPal-MMA) polymer membranes were prepared with a novel bi-solvent two-step phase-inversion method, and the schematic flow diagram of preparation for the membrane was shown in Figure 1. Firstly, the p(OPal-MMA) was dissolved in different mixed solvents at a solid-liquid ratio of 1:5 (g/mL). The composition of the mixed solvents and corresponding polymer weight was listed in Table 1. The obtained slurry was stirred at room temperature for 5 h and then uniformly coated onto the smooth glass pane with the spreader working in fume cupboards. Afterwards, the coated pane was moved to a deionized water-filled tray in order to remove the remaining soluble organic solvent. The submersed polymer membrane was then vacuum dried at 80 °C for 24 h: the thickness of the obtained dried membrane was 60 ± 5 μm. It was then punched into disks of 18 mm diameter by a slitting machine (MRX-CP60, Shenzhen mingruixiang, Shenzhen, China). Finally, the dried thin disk was immersed in 1 mol/L LiClO$_4$ solution (EC:PC = 1:1, *w/w*) for 2 h to obtain gel p(OPal-MMA) electrolytes.

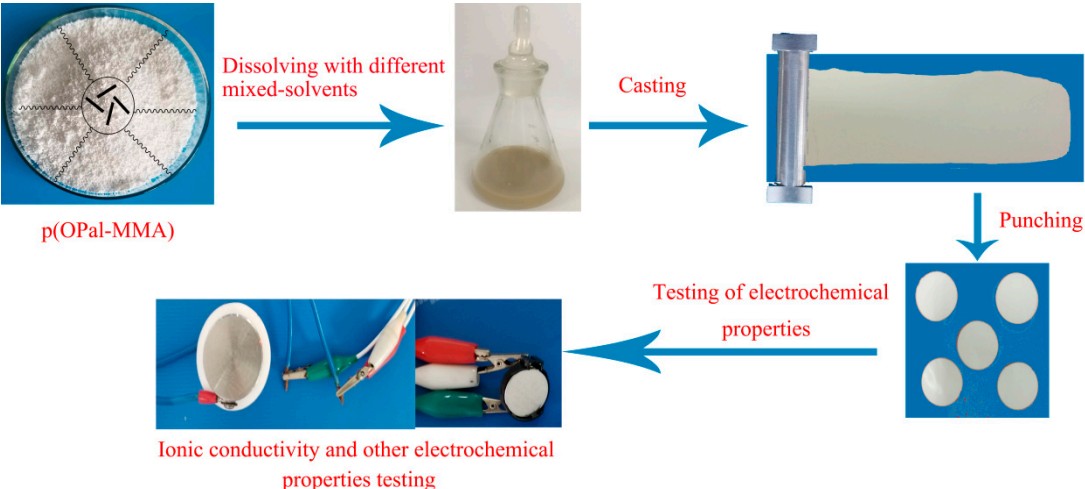

**Figure 1.** The preparation of p(OPal-MMA) polymer membrane.

**Table 1.** The amount of solvents and polymers for preparation of p(OPal-MMA) polymer membrane.

| Run | Solvent | | | | | | p(OPal-MMA) (g) |
|---|---|---|---|---|---|---|---|
| S1 | DMF/g | 5 | 6.25 | 7.5 | 8.75 | 10 | |
| | THF/g | 20 | 18.75 | 17.5 | 16.25 | 15 | |
| | Denoted | S1-1 | S1-2 | S1-3 | S1-4 | S1-5 | |
| S2 | DMF/g | 5 | 6.25 | 7.5 | 8.75 | 10 | |
| | Acetone/g | 20 | 18.75 | 17.5 | 16.25 | 15 | |
| | Denoted | S2-1 | S2-2 | S2-3 | S2-4 | S2-5 | 5 |
| S3 | DMAC/g | 2.5 | 3.75 | 5 | 6.25 | 7.5 | |
| | THF/g | 22.5 | 21.25 | 20 | 18.75 | 17.5 | |
| | Denoted | S3-1 | S3-2 | S3-3 | S3-4 | S3-5 | |
| S4 | DMAC/g | 2.5 | 3.75 | 5 | 6.25 | 7.5 | |
| | Acetone/g | 22.5 | 21.25 | 20 | 18.75 | 17.5 | |
| | Denoted | S4-1 | S4-2 | S4-3 | S4-4 | S4-5 | |

## 2.3. Characterization of Physical Properties

The porosity of the prepared p(OPal-MMA) membranes was determined with the *n*-butanol-absorbing method, as reported earlier [3,32]. The masses of the membranes were measured before and after immersion in *n*-butanol for 2 h, then porosity (%) was calculated according to the following Equation (1) [32]:

$$Porosity\ (\%) = \frac{M_{BuOH}/\rho_{BuOH}}{M_{BuOH}/\rho_{BuOH} + M_P/\rho_P} \times 100\% \tag{1}$$

where $M_P$ is the dry weight of the polymer membrane, $M_{BuOH}$ is the weight of the *n*-butanol sucked up, $\rho_{BuOH}$ and $\rho_P$ are the densities of the *n*-butanol and membrane, respectively.

## 2.4. Electrochemical Testing

Block coin-cells were assembled by sandwiching the gel p(OPal-MMA) electrolytes between two pieces of stainless steel (SS) blocking electrodes (surface area: 2.54 cm$^2$) and sealed in a glove box filled with Ar. Ionic conductivity was measured based on the AC impedance spectra with the frequency ranging from 0.1 Hz to 100 KHz at room temperature (25 °C) and then calculated by the Equation (2) as follows [33,34]:

$$\sigma = \frac{L}{R_b A} \tag{2}$$

where $R_b$, $L$ and $A$ are the bulk resistance (Ω), the thickness (μm) and the effective area (2.54 cm$^2$) of GPEs separately.

The lithium-ion transference number ($t_{Li+}$) was measured with the AC impedance technology and chronoamperometry method [35]. The Li/GPE/Li cells were assembled by sandwiching the GPEs between two pieces of lithium metal electrodes. The $t_{Li+}$ value is obtained from the following Equation (3):

$$t_{Li+} = \frac{I_s(\Delta V - I_0 R_0)}{I_0(\Delta V - I_0 R_0)} \tag{3}$$

where $\Delta V$ represents the polarization potential (10 mV); $I_0$ and $I_s$ are the initial current and steady-state current, while $R_0$ and $R_s$ are their corresponding interfacial resistances, respectively.

The electrochemical stability window of GPEs was measured by linear sweep voltammetry (LSV) at a voltage range of 2~6 V (vs. Li$^+$/Li) with a scanning rate of 0.5 mV/s of Li/GPE/SS on the electrochemical work station (CHI660E, Chenhua, Shanghai, China).

The LiFePO$_4$ electrode was prepared with the mixture of 80 wt% LiFePO$_4$ powder, 10 wt% PVDF and 10 wt% carbon black, and NMP was then used as a solvent to form the slurry. The slurries

were spread onto the aluminum foil with a spatula and dried at 80 °C for 48 h in vacuum, and the active material surface density of the dried cathode was about 1.30 mg/cm$^2$. The coin-cells of Li/GPE/LiFePO$_4$ with a lithium thickness of 1.0 mm were then assembled in a glove box filled with Ar. The charge-discharge performance of the Li/GPE/LiFePO$_4$ coin-cell was tested on a Neware test system (CT-4008-5V10mA-164, Shenzhen, China) over a potential window of 2.8 to 4.0 V.

## 3. Result

### 3.1. SEM Images of p(OPal-MMA) Membranes

The typical surface morphology of the membrane was observed by scanning electron microscopy (SEM); the corresponding results are shown in Figure 2. The S2-4 membrane has an interconnected framework and abundant pores. Besides, the pore size is uniformly distributed, which is not only beneficial for electrolyte retention but also provides more transmission channels for lithium ions. While the S1-3 membrane also has a porous structure, it is disordered with fewer pores. For membranes using DMAC as a porogen (S3-2 and S4-2), pores are rarely observed from the dense surface, indicating a relative low porosity, which is adverse to lithium ion transport.

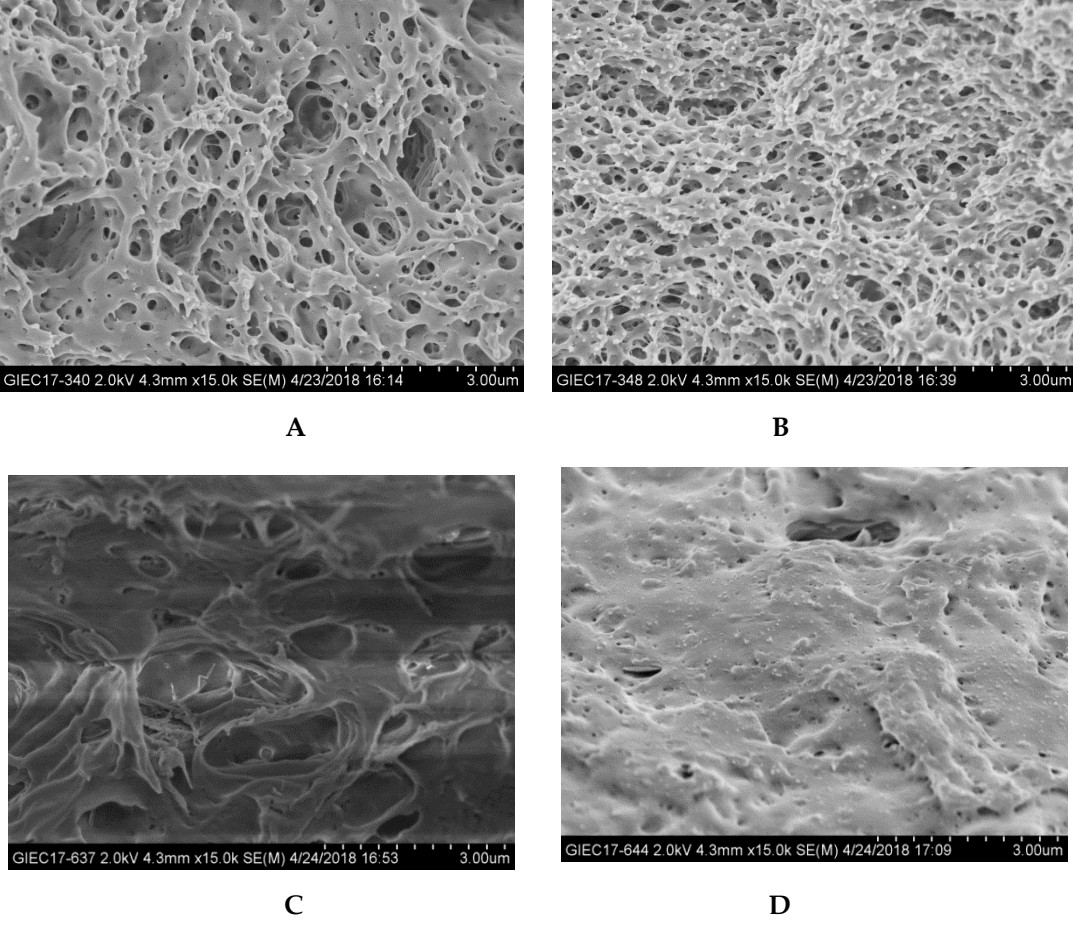

**Figure 2.** SEM images of the p(OPal-MMA) membranes prepared by different mixed solvents. (**A**), S1-3; (**B**), S2-4; (**C**), S3-2; (**D**), S4-2.

### 3.2. Porosity of the p(OPal-MMA) Membranes

In order to accurately determine the porosity of these membranes, the *n*-butanol-absorbing method was conducted for these membranes, as presented in Figure 2. The porosity of these membranes prepared by dissolved in different mixed solvents makes a great difference. When employing DMF

as porogen, the prepared membranes display a much higher porosity than that of DMAC. Especially when using acetone as the dissolvent under the same condition, the porosity can reach a maximum 42.6% (S2-4, Figure 3). These results could be mainly attributed to the faster vaporization rate of acetone and the higher water-solubility of DMF, which can promote the generation of more pores with uniform size.

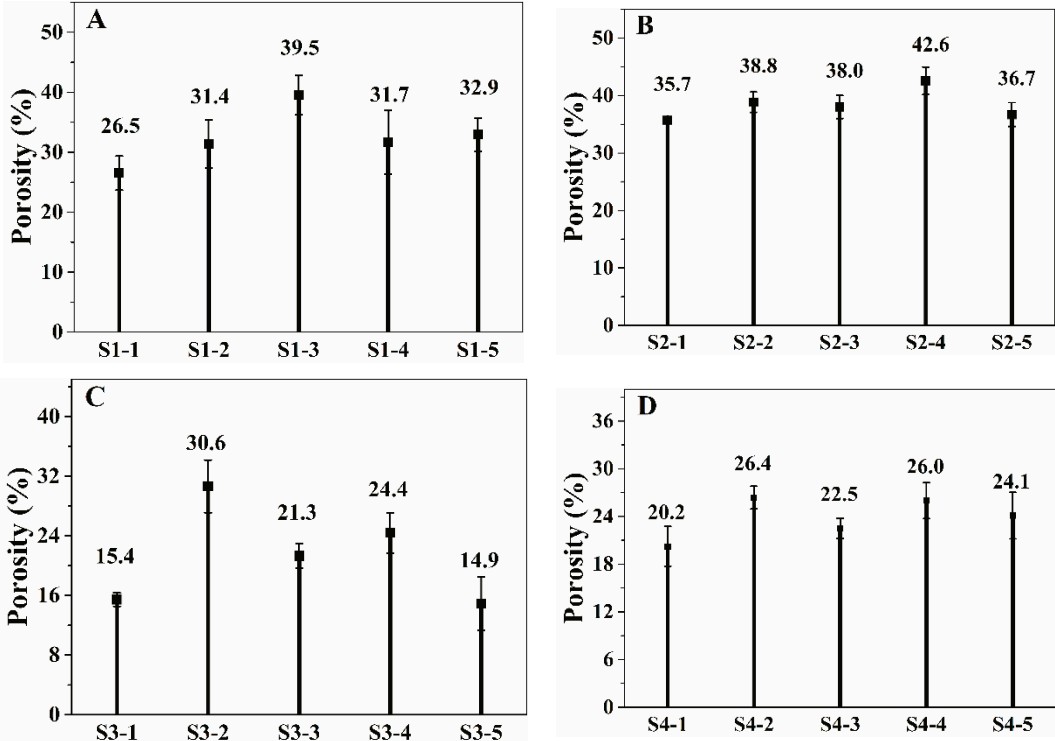

**Figure 3.** The porosity of the p(OPal-MMA) membranes prepared by different mix solvents. (**A**), S1; (**B**), S2; (**C**), S3; (**D**), S4.

### 3.3. The Ionic Conductivity of Gel p(OPal-MMA) Electrolyte

The ionic conductivity data of various GPEs prepared from membranes are shown in Figure 4. The ionic conductivity of all GPEs in this study was higher than $1.0 \times 10^{-3}$ S/cm. Generally, $1.0 \times 10^{-3}$ S/cm is the standard value of the ionic conductivity, determining whether the GPE has an application potential [3]. In addition, the GPE made of membrane S2-4 exhibited a more superior ionic conductivity than others without exception, even creating the highest ionic conductivity of GPE up to $3.99 \times 10^{-3}$ S/cm at room temperature.

### 3.4. The Ion Transference Number of Gel p(OPal-MMA) Electrolyte

Figure 5 shows the lithium ion transference numbers ($t_{Li+}$) of the GPEs. The lithium ion transference of liquid electrolyte cell with PE as separator was about 0.35 [36]. In contrast, the highest $t_{Li+}$ value in this work was 0.612 (as shown in Figure 5 S2-4), which was prepared by a mixed solvent of DMF and acetone. Notably, the lithium-ion transference number was related to ionic conductivity [37]. In this work, it can be seen that the GPE that displays a high ionic conductivity generally exhibits a high lithium-ion transference number, which may be related to the hydroxyl groups introduced by the Pal and the polar bond of –C=O in GPEs. The former can form hydrogen bonds with $ClO_4^-$ [38], while the latter can block the transport of large anions of $ClO_4^-$, thus promoting the lithium-ion transference. Moreover, the formation of the ion passing channels that Pal introduced and the porous structure of membranes both contribute to improving ionic conductivity and lithium ion transference.

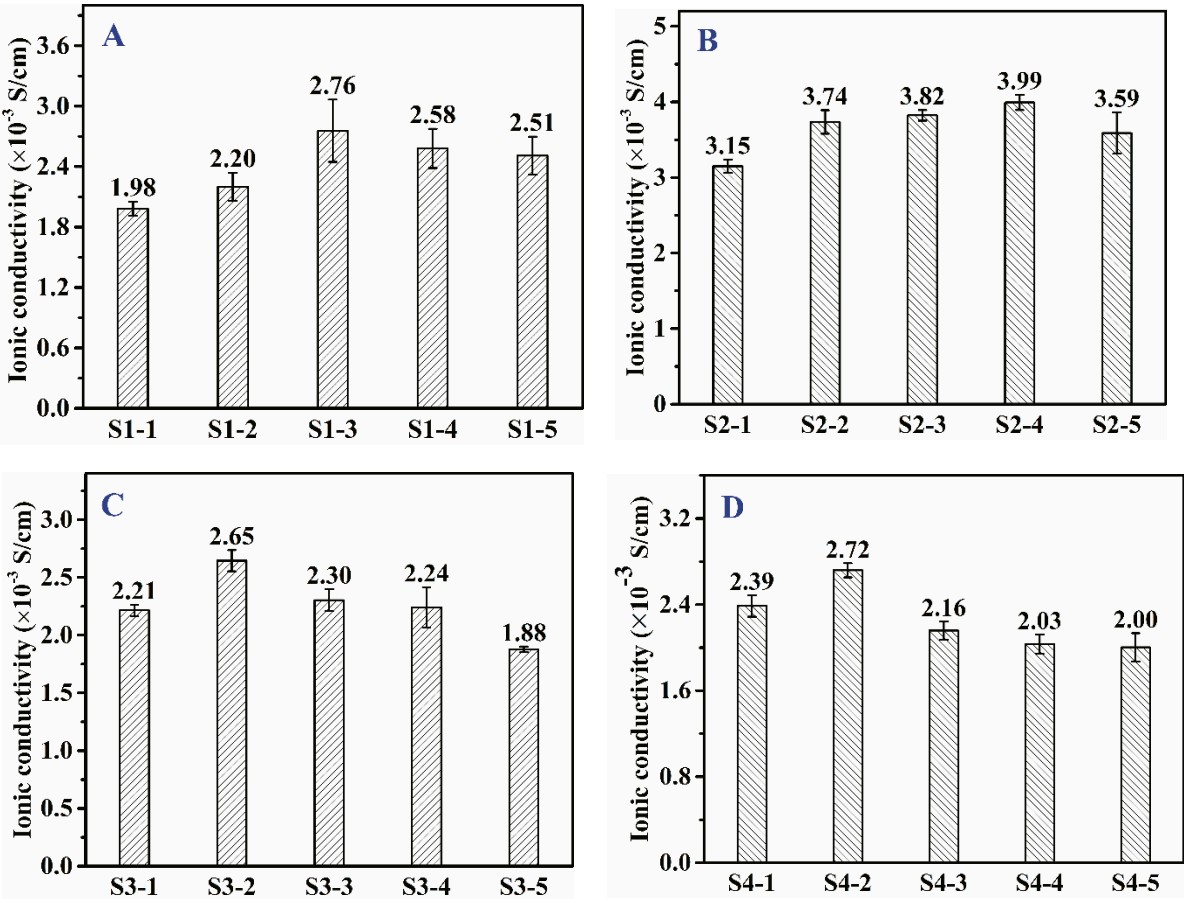

**Figure 4.** Ionic conductivities of GPEs prepared by different mixed solvents. (**A**), S1; (**B**), S2; (**C**), S3; (**D**), S4.

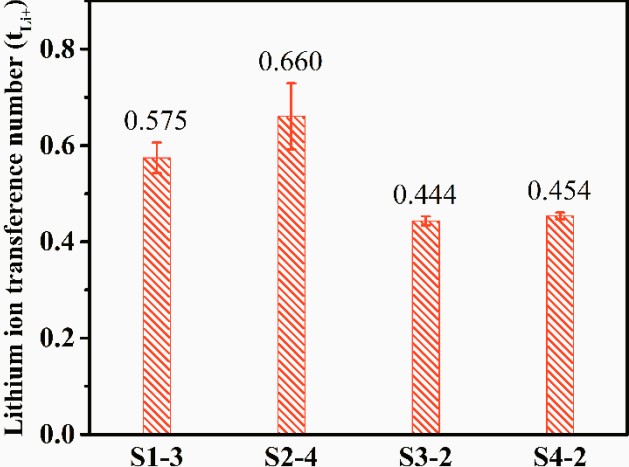

**Figure 5.** The lithium ion transference numbers of GPEs prepared by different mixing solvents.

*3.5. The Electrochemical Stability Window of Gel p(OPal-MMA) Electrolyte*

The electrochemical stability window of GPE is also a crucial indicator to evaluate its application feasibility in commercial LIBs. The LSV measurement of these four kinds of GPEs was conducted in a potential range of 2.0~6.0 V (vs. $Li^+/Li$) at a scan rate of 0.5 mV s$^{-1}$. As presented in Figure 6, there is little current flow below 4.5 V of all the Li/GPEs/SS cells, revealing that their electrochemical stability

window exceeds 4.5 V, which indicates that the obtained GPEs can meet the voltage requirements of practical market application in high-voltage LIBs.

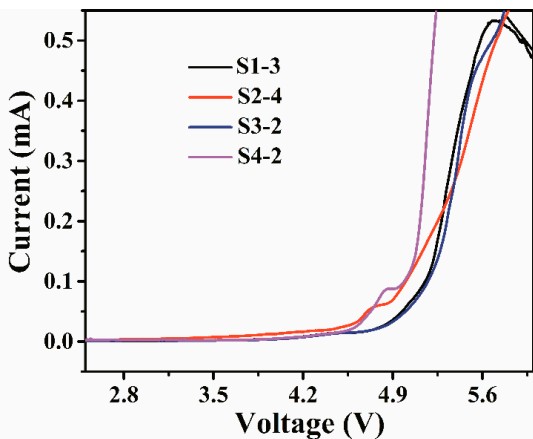

**Figure 6.** LSV curves of GPEs prepared by different mixed solvents.

### 3.6. Electrochemical Performance of Gel p(OPal-MMA) Electrolyte

In order to explore the practical application performance of GPEs in LIBs, the galvanostatic charge and discharge tests of Li/GPEs/LiFePO$_4$ cells were investigated and the initial profiles at 0.1 C rate were shown in Figure 6a. A typical charge-discharge potential plateau of LiFePO$_4$ appeared clearly at around 3.4~3.5 V, which was consistent with its typical electrochemical behavior previously reported [39–41], indicated a reversible cycling process. Remarkably, the GPE prepared from S2-4 membrane delivered an initial charge capacity of 167 mAh/g (Figure 7a); it was close to the theoretical specific capacity (170 mAh/g). The excellent electrochemical performance may be due to the high porosity and its consequence of outstanding ionic conductivity and considerable $t_{Li+}$ value. The cycling behavior experiment was also carried out with a constant current rate of 0.2 C from 2.8~4.0 V and the results are presented in Figure 7b. The GPE from S2-4 revealed the most excellent cycling performance: the cyclic discharge capacity was about 166~168 mAh g$^{-1}$ and there was almost no decline after 50 cycles. This superior cyclic performance could be attributed to the outstanding ionic conductivity, considerable $t_{Li+}$ number and good electrochemical interface stability of membranes. In addition, the coulombic efficiency of the GPE from S2-4 ranged from 96.74% to 99.98% at the whole cycle test, which reflected a good interfacial compatibility between the electrodes and the electrolyte [38]. The membrane could entrap more liquid electrolyte when it had more pores, which enabled membrane swelling and reduced interfacial resistance, and improved the interfacial stability between the GPE and electrode. Based on the above results, the GPE prepared with the DMF and acetone mixed solvent could have a good application prospect in LIBs because of its excellent reversible capacity and cyclic performance.

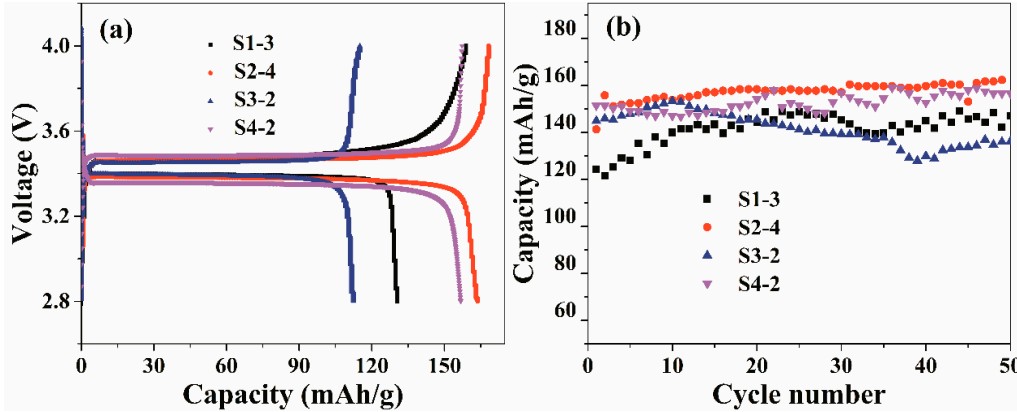

**Figure 7.** Initial charge-discharge capacities at 0.1 C rate (**a**) and cycling performances at 0.2 C rate (**b**) of GPEs prepared by different mixed solvents.

## 4. Conclusions

The p(OPal-MMA) membranes were prepared with the bi-solvent two-step phase-inversion method by using different kinds of mixed solvents. In comparison with other solvent combinations, when employing DMF as a porogen combined with acetone as the component solvent, the obtained membrane and GPE exhibited highest porosity (42%) and excellent electrochemical performance. The ionic conductivity and the lithium ion transference number can reach a high value of $3.99 \times 10^{-3}$ S/cm and 0.709 separately with the electrochemical stability window exceeding 4.5 V. When tested as Li/GPE/LiFePO$_4$ cells, it delivered the highest initial specific capacity of 167 mAh g$^{-1}$ at 0.1 C and retained most capacities after 50 cycles at 0.2 C. Overall, choosing an appropriate solvent plays a vital role in developing the GPEs, while the combination of DMF and acetone contributes to promoting the gel p(OPal-MMA) electrolyte as a potential separator applied in commercial LIBs.

**Author Contributions:** X.C. and H.Z. conceived and designed the experiments; L.T. and M.W. performed the experiments; H.G. and C.H. analyzed the data; X.C. contributed reagents/materials/analysis tools; L.T. wrote the original draft manuscript; H.Z revised the manuscript. All of the authors discussed the results in the manuscript.

**Funding:** This research was funded by the Project of Jiangsu Province Science and Technology (grant number: BE2016125), the tender project of HuaiAn (grant number: HAZC2018010036), and the Open Foundation of Xuyi Center of Attapulgite Applied Technology Research Development & Industrialization, Chinese Academy of Sciences (grant number: 201604).

**Conflicts of Interest:** The authors declare no conflict of interest.

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
