# Peer review of "The Effect of Different Mixed Organic Solvents on the Properties of p(OPal-MMA) Gel Electrolyte Membrane for Lithium Ion Batteries"

_applsci, doi:10.3390/app8122587_

Reviewer 1 Report

Tian et al. investigated the effects of mixed solvents on the performance of p(Opal-MMA) gels. The authors carefully adjusted the composition and combination of organic solvents, and the prepared p(Opal-MMA)s were analyzed in terms of surface morphology, porosity, ionic conductivity, electrochemical characteristics, and performance in Li-batteries. Overall, this study is well-organized and carefully conducted, but a few minor issues should be addressed before publication.

1. In introduction, PMMA is introduced as a polymer matrix for GPEs (line 44-50). However, recently various effective polymer matrices such as block copolymers and random copolymers have been reported for achieving high performance GPEs. This should be updated in introduction with references below.

For PS-b-PMMA-b-PS triblock copolymers: a) J. Am. Chem. Soc. 2014, 136, 3705 ; b) Chem. Mater. 2015, 27, 1420.

For poly[styreneran1(4vinylbenzyl)3methylimidazolium hexafluorophosphate] (P[SrVBMI][PF6]): Adv. Funct. Mater. 2018, 28, 1706948

For P(VDF-co-HFP): a) J. Mater. Chem. C 2016, 4, 8448; b) ACS Appl. Mater. Interfaces 2017, 9, 7658.

2. The authors have reported P(Opal-PMMA) and their applications in GPEs (ref 12; line 55-58) previously. If the authors make clear a difference between the previous work (ref 12) and current work, the contents in this work can be emphasized.

3. In section 2.3, the authors employed the n-butanol-absorbing method to estimate a porosity of P(Opal-PMMA) membranes. The component in the polymer, particularly PMMA, may not be compatible with n-butanol. Therefore, I anticipate that the adsorption of n-butanol into the pores is not preferred and empty pores may be present even during the test. Why did the authors use n-butanol for the test?

4. Line 118-119: MP and MBuOH should be expressed by subscripts.

5. Line 162-168: The authors claimed that the higher porosity can be induced when a mixture of low bp acetone (showing fast evaporation) and DMF (having higher water solubility) was employed. I agree that the faster evaporation rate of acetone is likely to contribute forming highly porous structures. However, the function of DMF is still unclear. Why high water solubility of DMF is important for this case?

Author Response

Dear Editor/Reviewer:

Thank you for your letter and the reviewers’ comments concerning our manuscript entitled “The effect of different mixed organic solvents on the properties of p(OPal-MMA) gel electrolyte membrane for lithium ion batteries” (399764). Those comments are all valuable and very helpful for revising and improving our paper, as well as the important guiding significance to our researches. We have modified the manuscript accordingly, and the detailed corrections are using the "Track Changes" of the Word. The main corrections are listed below point by point:

We revised the whole manuscript carefully. Thank reviewers for his valuable guideline, and I have learned a lot.

With many thanks for your cordially help.

 Yours sincerely,

Xinde Chen

Response to Reviewer 1 Comments

 Point 1. In introduction, PMMA is introduced as a polymer matrix for GPEs (line 44-50). However, recently various effective polymer matrices such as block copolymers and random copolymers have been reported for achieving high performance GPEs. This should be updated in introduction with references below.

 For PS-b-PMMA-b-PS triblock copolymers: a) J. Am. Chem. Soc. 2014, 136, 3705 ; b) Chem. Mater. 2015, 27, 1420.

 For poly[styrene‐ran‐1‐(4‐vinylbenzyl)‐3‐methylimidazolium hexafluorophosphate] (P[S‐r‐VBMI][PF6]): Adv. Funct. Mater. 2018, 28, 1706948

 For P(VDF-co-HFP): a) J. Mater. Chem. C 2016, 4, 8448; b) ACS Appl. Mater. Interfaces 2017, 9, 7658.

 Response 1: Thanks for your comments. We have read these literatures carefully and cited them in the INTRODUCTION of the revised manuscript.

 Point 2. The authors have reported P(Opal-PMMA) and their applications in GPEs (ref 12; line 55-58) previously. If the authors make clear a difference between the previous work (ref 12) and current work, the contents in this work can be emphasized.

 Response 2: Thanks for your comments. Previous study focus on the polymerization and characterization of p(OPal-PMMA) polymer, while this study mainly investigate the preparation technology of the polymer membrane by the phase inversion method with complex solvent. The main difference was added in the INTRODUCTION of the revised manuscript.

Point 3. In section 2.3, the authors employed the n-butanol-absorbing method to estimate a porosity of p(OPal-PMMA) membranes. The component in the polymer, particularly PMMA, may not be compatible with n-butanol. Therefore, I anticipate that the adsorption of n-butanol into the pores is not preferred and empty pores may be present even during the test. Why did the authors use n-butanol for the test?

 Response 3: Thanks for your comments. The surface tension and contact angle of the n-butanol are closed to liquid electrolyte, thus the polymer membrane can absorb almost equal amounts of n-butanol compare to that of liquid electrolyte. Many literatures have used n-butanol to estimate a porosity of polymer membrane. (Journal of Power Sources 160 (2006) 1320–1328, Electrochimica Acta 58 (2011) 674– 680, Journal of Power Sources 201 (2012) 294-300, Ionics (2012) 18:47–53, Journal of Membrane Science 444(2013)213–222, Journal of Power Sources 307 (2016) 320-328).

 Point 4. Line 118-119: MP and MBuOH should be expressed by subscripts.

 Response 4: Thanks for your comments. The words of “MP” and “MBuOH” have been replaced by “Mp” and “MBuOH”.

 Point 5. Line 162-168: The authors claimed that the higher porosity can be induced when a mixture of low bp acetone (showing fast evaporation) and DMF (having higher water solubility) was employed. I agree that the faster evaporation rate of acetone is likely to contribute forming highly porous structures. However, the function of DMF is still unclear. Why high water solubility of DMF is important for this case?

 Response 5: Thanks for your comments. When the wet membrane was rapidly immersed in the aqueous phase, phase separation will be take place due to the difference of the volatility and dissolvability of the solvent, polymer and water. The high water solubility of DMF can guarantee its easily and completely remove from the membrane by water, which will bring about more pores in the membrane.

Reviewer 2 Report

Submitted manuscript describes an attempt of the development of new gel polymer electrolytes for lithium ion batteries cell that is important and relevant task toward next generation high performing batteries. The overall quality of the manuscript is acceptable for publishing in Applied Science. However, from my modest point of view, some points must be improved before its acceptance.

Line 107. Please specify thickness range of the prepared membranes.

Line 130. Please clarify whether separator or GPE was used for tLi+ measurements.

Lines 139-144. Please specify loading of positive electrode and thickness of Li anode. Also, I recommend to rewrite the paragraph in chronological order starting from the cathode preparation then describing the cell assembly and, finally, explaining cell cycling conditions.

Figure 2. It is not clear which membranes are titled as S1, S2 etc. because in Table 2 authors described that there are 4 membranes (S1-1, S1-2 etc.) of each family. Please clarify this point.

Line 132. At which temperature the ionic conductivity values were measured?

Table 2. It is not clear which membranes are titled as S1, S2 etc. because in Table 2 authors described that there are 4 membranes (S1-1, S1-2 etc.) of each family. Please clarify this point. In addition, I recommend specifying standard deviation of tli+ measurement and adding corresponding value for used liquid electrolyte as reference.

Fig. 5. I think in all presented inset figures steady state current has not been reached. Therefore, tli+ data are not correct and must be removed from the manuscript or replaced by correct ones.

Lines 200-201. In order prove you statement about compatibility of the prepared electrolytes and high voltage cathodes real experimental data must be presented because it is well known that CV is always overestimates the electrochemical stability range of solid electrolytes [Han, F.; Zhu, Y.; He, X.; Mo, Y.; Wang C. Electrochemical Stability of Li10GeP2S12 and Li7La3Zr2O12 Solid Electrolytes. Adv. Energy Mater. 2016, 1501590. DOI: 10.1002/aenm.201501590].

Line 205. Strictly speaking, it is not a polarization curve. Please use correct electrochemical terminology.

Figure 6. It is not clear which membranes are titled as S1, S2 etc. because in Table 2 authors described that there are 4 membranes (S1-1, S1-2 etc.) of each family. Please clarify the point.

Lines 210, 217 and 232. Please check which C-rate was used: 0.1C or 0.2C?

Line 222-223. I think coulombic efficiency about 96% is quite low value for LiFePO4 electrode. Therefore, I cannot agree with your statement in 223 line.

Figure 7. It is not clear which membranes are titled as S1, S2 etc. because in Table 2 authors described that there are 4 membranes (S1-1, S1-2 etc.) of each family. Please clarify this point.

Figure 7a -7b. Why sample S3 has delivered<110 mAh/g on first cycle (Fig.7a) whereas in Fig.7b the same cell has shown about 150 mAh/g on the first cycling cycle. Please check and explain.

Figure 7b. Authors should explain why discharge capacity of samples S1, S2 and S4 are increasing over cycling that is not typical for Li-LiFePO4 chemistry.

Lines 225-228 & Figure 8. I think the explanation and figure should be improved toward better understanding by a reader. Moreover, additional experiments or simulations are needed to prove suggested hypothesis.

Line 280 – ref. 13. Please check surnames of all authors.

Author Response

Dear Editor/Reviewer:

Thank you for your letter and the reviewers’ comments concerning our manuscript entitled “The effect of different mixed organic solvents on the properties of p(OPal-MMA) gel electrolyte membrane for lithium ion batteries” (399764). Those comments are all valuable and very helpful for revising and improving our paper, as well as the important guiding significance to our researches. We have modified the manuscript accordingly, and the detailed corrections are using the "Track Changes" of the Word. The main corrections are listed below point by point:

We revised the whole manuscript carefully. Thank reviewers for his valuable guideline, and I have learned a lot.

With many thanks for your cordially help.

 Yours sincerely,

Xinde Chen

Response to Reviewer 2 Comments

 Point 1 Line 107. Please specify thickness range of the prepared membranes.

 Response 1: Thanks for your comments. The thickness of the membrane was about 60±5 μm, which was added into the revised manuscript.

 Point 2 Line 130. Please clarify whether separator or GPE was used for tLi+ measurements.

 Response 2: Thanks for your comments. The “separator” has been replaced by “GPE” in the revised manuscript..

 Point 3 Lines 139-144. Please specify loading of positive electrode and thickness of Li anode. Also, I recommend to rewrite the paragraph in chronological order starting from the cathode preparation then describing the cell assembly and, finally, explaining cell cycling conditions.

 Response 3: Thanks for your comments. The thickness of lithium is 1.0 mm and the loading of positive electrode is about 1.30 mg/cm2. This paragraph has been rewritten according to your suggestion in the revised manuscript.

 Point 4 Figure 2. It is not clear which membranes are titled as S1, S2 etc. because in Table 2 authors described that there are 4 membranes (S1-1, S1-2 etc.) of each family. Please clarify this point.

 Response 4: Thanks for your comments. “S1, S2, S3 and S4” have been replaced by “S1-3, S2-4, S3-2 and S4-2” respectively in Figure 2.

 Point 5 Line 132. At which temperature the ionic conductivity values were measured?

 Response 5: Thanks for your comments. The ionic conductivity was measured at“ 25 °C, which was added into the revised manuscript.

 Point 6 Table 2. It is not clear which membranes are titled as S1, S2 etc. because in Table 2 authors described that there are 4 membranes (S1-1, S1-2 etc.) of each family. Please clarify this point. In addition, I recommend specifying standard deviation of tli+ measurement and adding corresponding value for used liquid electrolyte as reference.

 Response 6: Thanks for your comments. The lithium ion transference of liquid electrolyte cell with PE as separator was about 0.35 (ACS Appl. Mater. Interfaces 2016, 8, 32637-32642), and the highest tLi+ value in this work was 0.612 (S2-4), which was prepared by a mixed solvent of DMF and acetone. Furthermore, the Table 2 has been replaced by Fig.5, and the “S1, S2, S3 and S4” have been replaced by “S1-3, S2-4, S3-2 and S4-2” respectively in Figure 5.

 Point 7 Fig. 5. I think in all presented inset figures steady state current has not been reached. Therefore, tli+ data are not correct and must be removed from the manuscript or replaced by correct ones.

 Response 7: Thanks for your comments. This section experiment has been renewed and the results were shown in Fig. 5 in the revised manuscript.

 Point 8 Lines 200-201. In order prove you statement about compatibility of the prepared electrolytes and high voltage cathodes real experimental data must be presented because it is well known that CV is always overestimates the electrochemical stability range of solid electrolytes [Han, F.; Zhu, Y.; He, X.; Mo, Y.; Wang C. Electrochemical Stability of Li10GeP2S12 and Li7La3Zr2O12 Solid Electrolytes. Adv. Energy Mater. 2016, 1501590. DOI: 10.1002/aenm.201501590].

 Response 8: Thanks for your comments. This section experiment has been rewritten in the revised manuscript

 Point 9 Line 205. Strictly speaking, it is not a polarization curve. Please use correct electrochemical terminology.

 Response 9: Thanks for your comments. This section experiment has been renewed and the results were shown in Fig. 5 in the revised manuscript.

 Point 10 Figure 6. It is not clear which membranes are titled as S1, S2 etc. because in Table 2 authors described that there are 4 membranes (S1-1, S1-2 etc.) of each family. Please clarify the point.

 Response 10: Thanks for your comments. “S1, S2, S3 and S4” have been replaced by “S1-3, S2-4, S3-2 and S4-2” respectively in Figure 6.

 Point 11 Lines 210, 217 and 232. Please check which C-rate was used: 0.1C or 0.2C?

 Response 11: Thanks for your comments. Initial charge-discharge was test at 0.1 C and charge-discharge cycles were measured at 0.2 C. “Initial charge-discharge capacities (a) and cycling performances (b) of GPEs prepared by different mixed solvents at 0.1 C” has been replaced by “Initial charge-discharge capacities at 0.1 C rate (a) and cycling performances at 0.2 C rate (b) of GPEs prepared by different mixed solvents”.

 Point 12 Line 222-223. I think coulombic efficiency about 96% is quite low value for LiFePO4 electrode. Therefore, I cannot agree with your statement in 223 line.

 Response 12: Thanks for your comments. The coulombic efficiency of the GPE from S2-4 was range from 96.74% to 99.98% at the whole cycle test.

 Point 13 Figure 7. It is not clear which membranes are titled as S1, S2 etc. because in Table 2 authors described that there are 4 membranes (S1-1, S1-2 etc.) of each family. Please clarify this point.

 Response 13: Thanks for your comments. “S1, S2, S3 and S4” have been replaced by “S1-3, S2-4, S3-2 and S4-2” respectively in Figure 7.

 Point 14 Figure 7a -7b. Why sample S3 has delivered<110 mAh/g on first cycle (Fig.7a) whereas in Fig.7b the same cell has shown about 150 mAh/g on the first cycling cycle. Please check and explain.

Response 14: Thanks for your comments. The less porosity of S3 was not beneficial to migration of lithium ion, thus its initial charge-discharge capacity is lower as well. Even while the discharge capacity of S3 near to 150 mAh/g around 10 cycles in Figure 7b, which mainly ascribed to the full activation of cathode. However, after 10 cycles the discharge capacity shows a sharp decline trend and presents a serious fluctuation during charge-discharge cycles. Furthermore, the initial charge-discharge capacities were tested at 0.1 C rate while the cycling performances were tested at 0.2 C, thus the data were different between them.

 Point 15 Figure 7b. Authors should explain why discharge capacity of samples S1, S2 and S4 are increasing over cycling that is not typical for Li-LiFePO4 chemistry.

Response 15: Thanks for your comments. This phenomenon showed in Fig. 7b was mainly ascribed to the activate process of cathode (LiFePO4) during charge-discharge cycles.

 Point 16 Lines 225-228 & Figure 8. I think the explanation and figure should be improved toward better understanding by a reader. Moreover, additional experiments or simulations are needed to prove suggested hypothesis.

 Response 16: Thanks for your comments. Unfortunately, due to the time reason, we cannot complete the additional experiments in one week, so the Figure 8 was deleted in the revised manuscript, and these works will be complete in the next work.

 Point 17 Line 280 – ref. 13. Please check surnames of all authors.

 Response 17: Thanks for your comments. All authors’ surnames of the Ref. 13 have been checked and they were correct.

Round  2

Reviewer 1 Report

The authors well addressed my concerns. Therefore, I recommend the publication of this work in this journal.

Author Response

Dear Reviewer:

Thank you for your valuable comments concerning our manuscript. The comments are all valuable and very helpful for our further researches. Most of all, your rigorous spirit of scientific research and exploration is worthy of admiration.

With hearty gratitude for your cordially advice and help.

 Yours sincerely,

Xinde Chen

Reviewer 2 Report

I think authors significantly improved the quality of the manuscript. However, from my modest point of view, few points still must be improved before its acceptance by the journal.

Line 215. Strictly speaking, it is not a polarization curve. Please use correct electrochemical terminology as it has been mentioned in 1st review.

Fig. 5. Inset figures of the Fig.5 in 1st and 2nd manuscript versions are quite different and look quite strange for me. Therefore, I can conclude that tli+ data are not correct and must be removed from the manuscript or replaced by correct ones. Again, I strongly recommend specifying the standard deviation of tli+ measurement to see the reproducibility.

Author Response

Dear Reviewer:

Thank you for your comments concerning our manuscript entitled “The effect of different mixed organic solvents on the properties of p(OPal-MMA) gel electrolyte membrane for lithium ion batteries” (399764) very much. Those comments are all valuable and very helpful for revising and improving our paper. We have modified the manuscript according to your valuable feedback, and the detailed corrections are using the "Track Changes" of the Word. The main corrections are listed below point by point. In addition, your rigorous spirit of scientific research and exploration is very worthy of admiration.

With many thanks for your cordially advice and help.

Yours sincerely,

Xinde Chen

Point 1 Line 215. Strictly speaking, it is not a polarization curve. Please use correct electrochemical terminology as it has been mentioned in 1st review.

 Response 1: Thanks for your comments. The “polarization curve” should be “chronoamperometric curve” and it has been deleted in the revised manuscript due to the Fig. 5 was changed.

 Point 2 Fig. 5. Inset figures of the Fig.5 in 1st and 2nd manuscript versions are quite different and look quite strange for me. Therefore, I can conclude that tli+ data are not correct and must be removed from the manuscript or replaced by correct ones. Again, I strongly recommend specifying the standard deviation of tli+ measurement to see the reproducibility.

 Response 2: Thanks for your comments. We have measured the tLi+ again after you pointed out in 1st review. There was a little difference of tLi+ between 1st and 2nd manuscript which mainly could be ascribed to experimental error. The value of tLi+ and the standard deviation of tLi+ have been shown in Fig.5 in revised manuscript.

Round  3

Reviewer 2 Report

I think the quality of the manuscript is acceptable for its publishing.